# Quantifying the Influence of Climate Change and Anthropogenic Activities on the Net Primary Productivity of China's Grasslands

**Xiafei Zhou [1,2], Binbin Peng [1], Ying Zhou [2], Fang Yu [2,\*] and Xue-Chao Wang [3,4,\*]**

1 College of Management and Economics, Tianjin University, Tianjin 300072, China
2 The Center for Eco-Environmental Accounting, Chinese Academy of Environmental Planning, Beijing 100012, China
3 State Key Laboratory of Earth Surface Processes and Resource Ecology, Faculty of Geographical Science, Beijing Normal University, Beijing 100875, China
4 School of Natural Resources, Faculty of Geographical Science, Beijing Normal University, Beijing 100875, China
\* Correspondence: yufang@caep.org.cn (F.Y.); xcwang@bnu.edu.cn (X.-C.W.)

**Abstract:** As one of China's most common vegetation types, grasslands comprise about 27.5% of its terrestrial area and 41% of its carbon storage. Since climate change (CC) and human activities (HA) have a great effect on grasslands, quantifying the contributions of CC and HA on grassland net primary productivity (NPP) is crucial in understanding the mechanisms of grassland regional carbon balances. However, current approaches, including residual trend, biophysical model and environmental background-based methods, have limitations on different scales, especially on the national scale of China. To improve assessment accuracy, modifications to the environmental background-based method were introduced in calculating the CC and HA contributions to the actual NPP (ANPP). In this study, the grassland ANPP in national nature reserves was defined as the environmental background value (PNPP), which was only affected by CC and without HA. The pixel PNPP outside the nature reserves could be replaced by the pixel PNPP in the nature reserve with the most similar habitat in the same natural ecological geographical division. The impact of HA on grassland ANPP (HNPP) could be identified by calculating the difference between PNPP and ANPP. Finally, the contributions of CC and HA to ANPP changes were assessed by the trends of ANPP, PNPP, and HNPP. The results showed that the average grassland ANPP significantly increased from 2001 to 2020. CC contributed 71.0% to ANPP change, whereas HA contributed 29.0%. Precipitation was the main contributor to grassland growth among arid and semi-arid regions, while temperature inhibited productivity in these areas. HA was the major cause of degradation in China's grasslands, although the effects have declined over time. The research could provide support support for government decisions. It could also provide a new and feasible research method for quantitatively evaluating grasslands and other ecosystems.

**Keywords:** grassland; NPP; climate change (CC); human activities (HA); contribution

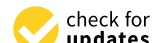


## 1. Introduction

As an essential component in the ecosystem, vegetation is indispensable in maintaining climatic stability and carbon balance and is crucial in achieving carbon emissions neutrality targets [1–3]. Net primary productivity (NPP) is the net carbon sequestered via vegetation photosynthesis in unit time and unit area [4,5]. It characterizes the carbon sequestration capacity of vegetation and serves as a vital gauge of plant dynamics [6–8]. Climate change (CC) and human activities (HA) are two main influencing aspects of changes in vegetation NPP [9,10]. CC (e.g., precipitation and temperature change) significantly affects the intensity of vegetation photosynthesis and respiration [11,12], leading to an

increase or decrease in vegetation NPP [7,13]. HA, such as overgrazing or land use change, may result in severe ecological degradation [9,14]. Therefore, estimating the influence of CC and HA on vegetation NPP is crucial for vegetation management.

Recent studies have investigated the influence of CC and HA on vegetation NPP [1,9,12]. Currently, the most widely used methods mainly include regression analysis methods [15,16], residual trend methods [1,17], biophysical model methods [9,10] and environmental background-based methods [18]. The regression analysis method is relatively simple and directly uses the regression model to quantify the impact of CC and HA on vegetation NPP [19]. However, this approach has difficulty identifying the respective influence of CC and HA on vegetation NPP [20]. The residual trend method indirectly evaluates the effects of anthropogenic activities on vegetation NPP by simulating the residual trend of actual NPP and potential NPP [1,21]. However, this method assumes all vegetation is affected by CC and HA, resulting in deviations from the actual situation [22,23]. For example, HA has no impact on vegetation in national nature reserves [18]. The biophysical model method employs the ThornThwaite Memorial Model and Carnegie-Ames-Stanford Approach in estimating the potential and actual NPP values and then separating the relative contributions of CC and HA in vegetation NPP [24,25]. However, numerous parameters evaluating vegetation physiology and ecology would have to be measured, thus increasing the model uncertainties [10,26]. The environmental background-based method is a new way to separate the impacts of CC and HA on vegetation NPP [18]. In this method, the vegetation NPP in nature reserves is defined as the environmental background value, which is affected only by CC and without HA interference. The impact of HA on vegetation NPP is then characterized by calculating the difference between the environmental background value and the NPP outside the reserve. This method provides a more precise approach but ignores the influence of diverse habitat conditions on vegetation NPP. Improvements and modifications to the environmental background-based approach have to be incorporated to generate more accurate estimates of the influence of CC and HA on vegetation NPP.

Grassland is a major ecosystem in China, accounting for about 27.5% of China's total land area [27,28]. As a carbon sink, grassland ecosystems play a critical role and represent about 41% of China's carbon storage [29,30]. However, due to climate warming, land-use changes, as well as overgrazing, many grasslands in China have suffered serious ecological degradation [10,31], causing significant decline in the regional carbon sequestration capacity [32,33]. Grassland degradation has also resulted in a number of environmental problems (e.g., soil erosion and sandstorms) [34,35]. Since the late 1990s, the Chinese government has implemented policies to restore degraded grasslands, such as the Grain-for-Green Project launched in 1999 and Regulations on Conversion of Farmland to Forests in 2016 [31,36]. Succeeding environmental policies and restoration strategies would require a more accurate estimation of the climatic and anthropogenic impacts on grassland NPP, not merely as an academic exercise, but to provide important references for grassland management [10,37].

Against this background, an improved environmental background-based method was developed in this research to identify the key influencing factors of grassland NPP change in China. The objectives of this study were as follows: (i) investigate spatiotemporal dynamics of grassland NPP from 2001 to 2020, (ii) quantify the relative contributions of CC and HA on grassland NPP, and (iii) explore the relationships between grassland NPP and major climate factors.

## 2. Materials and Methods

### 2.1. Study Area

China's grassland ecosystem was chosen as the study area, distributed in the administrative regions of Gansu, Shaanxi, Xinjiang, Tibet, Inner Mongolia, and Hebei (Figure 1) [38–40]. These areas are primarily located in arid and semi-arid regions [39]. In terms of overgrazing and land-use change over the years, the ecological degradation in the study area is becoming more and more serious [41,42].

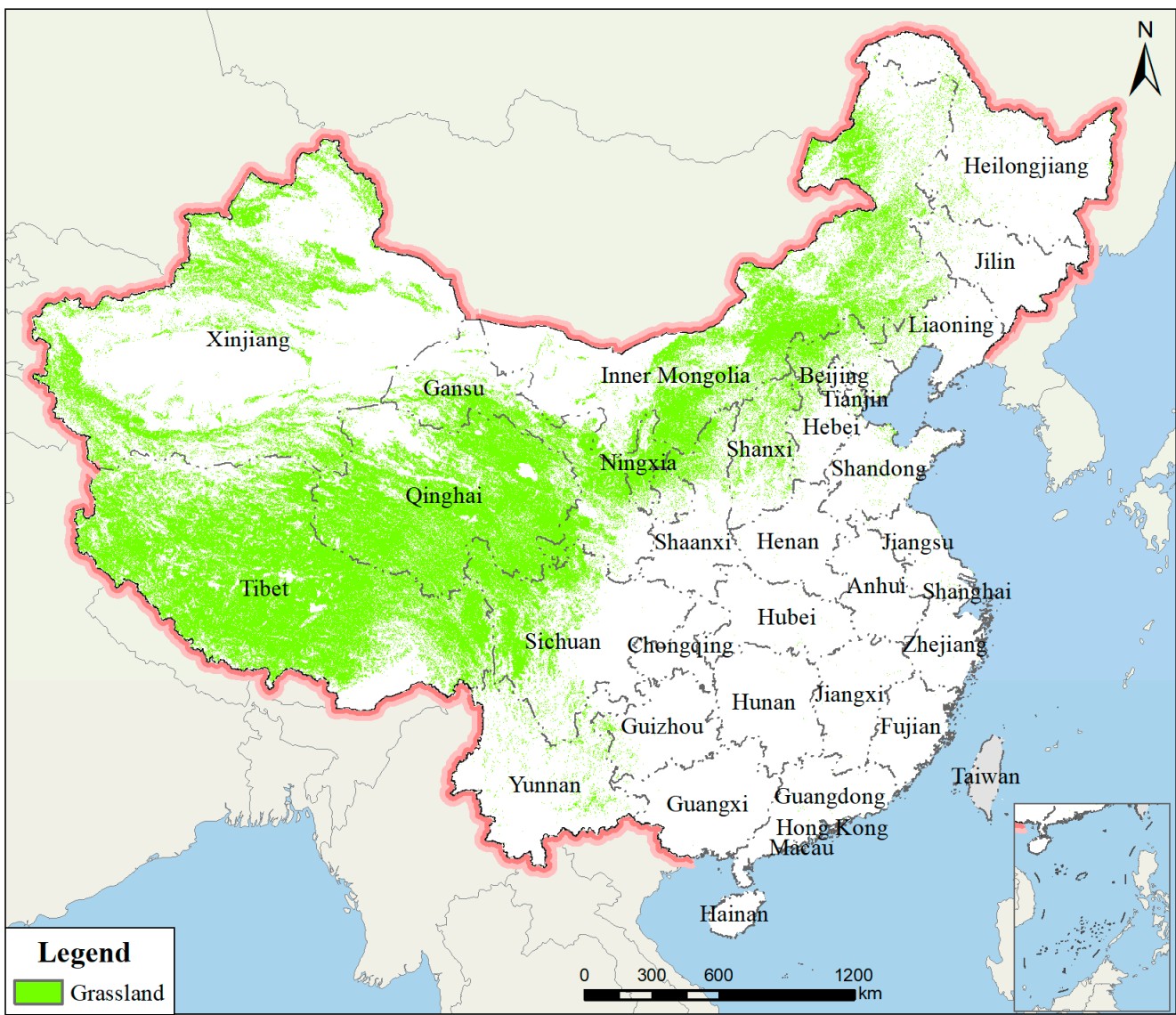

**Figure 1.** Study area.

### 2.2. Data Sources

The main research data was composed of land use, net primary productivity, precipitation, temperature, eco-geographical zoning, and national natural reserves data. The land use data for 2000–2018, the net primary productivity data for 2001–2020 and precipitation and temperature during 2001–2020 were obtained from the European Space Agency, the United States Geological Survey and the China Meteorological Data Network, respectively. The eco-geographical zoning data and DEM data were obtained from the Resource and Environment Science and Data Center (Figure 2). The national natural reserves data was derived from China's Ministry of Ecology and Environment (Figure 3) and is separated into three zones: core area, buffer zone and experimental area [43]. The core area is strictly protected from any human activity, but in the buffer zone, scientific research and observations are allowed. In the experimental area, visits and investigations, tourism, reproduction of rare and endangered wild animals and plants, and specific production activities are allowed [44]. All data were resampled to a spatial resolution of 1 km×1 km for unified analysis (Table 1).

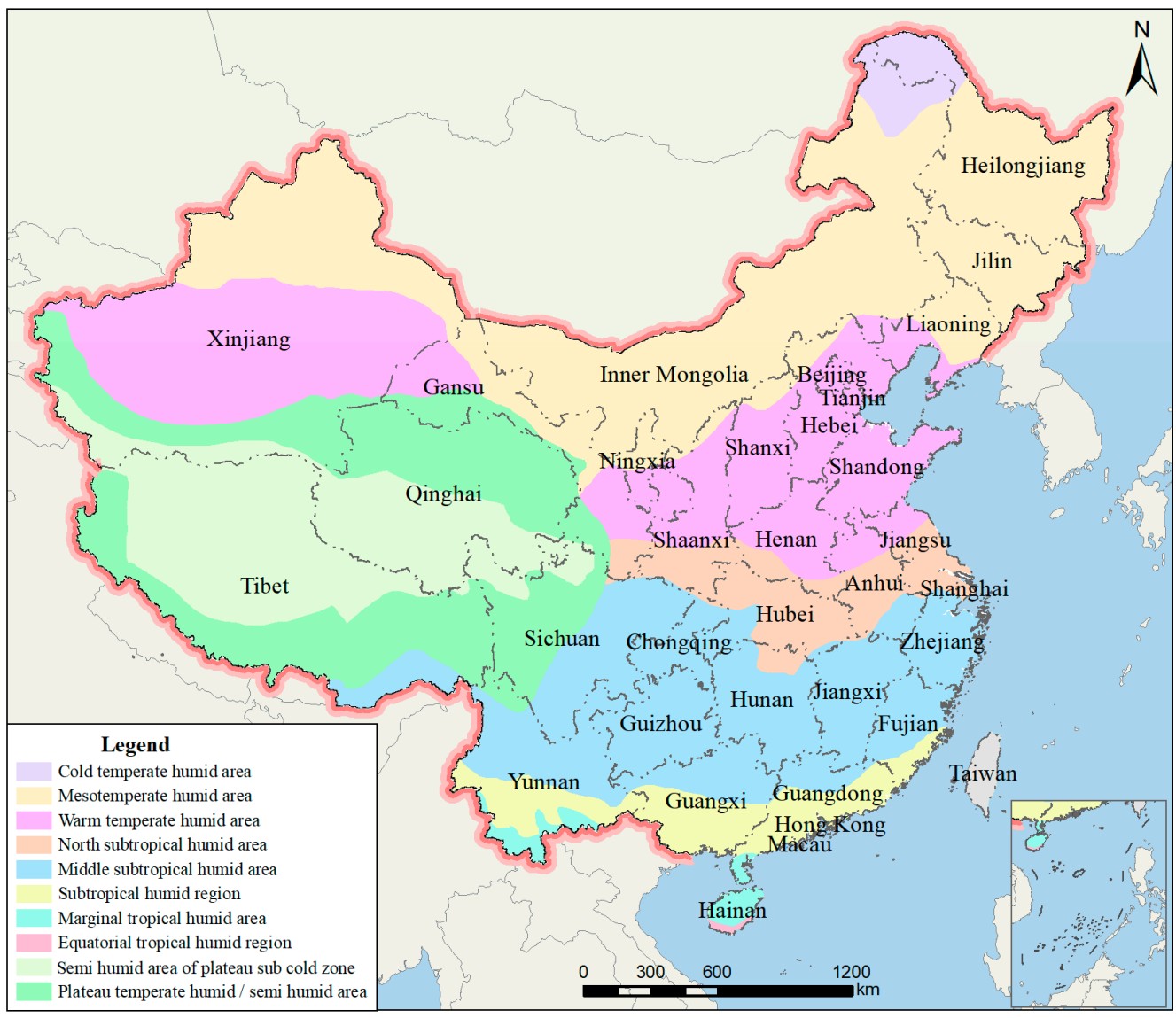

**Figure 2.** Eco-geographical zoning data in China.

**Table 1.** Data sources.

| Data | Website | Data Type | Resolution |
|---|---|---|---|
| LULC (CCI land cover) | https://www.esa-landcover-cci.org (4 January 2022) | raster | 300 m |
| NPP | https://lpdaacsvc.cr.usgs.gov (10 January 2022) | raster | 1 km |
| Precipitation and temperature | https://data.cma.cn/ (16 January 2022) | vector | |
| Eco-geographical zoning | https://www.resdc.cn/ (25 January 2022) | vector | |
| DEM | https://www.resdc.cn/ (27 January 2022) | raster | 1 km |
| National natural reserves | https://www.resdc.cn/ (28 January 2022) | vector | |

*2.3. Methods*

2.3.1. Extracting the Spatial Range of Grassland

Due to continuous changes in the grassland ecosystem from 2001 to 2020, only the unaltered grassland areas were considered in the study. The unchanged grassland areas in 2001, 2006, 2011, 2016, and 2018 were used as the final spatial grassland range (Figure 4).

**Figure 3.** The spatial distribution of national nature reserves in China.

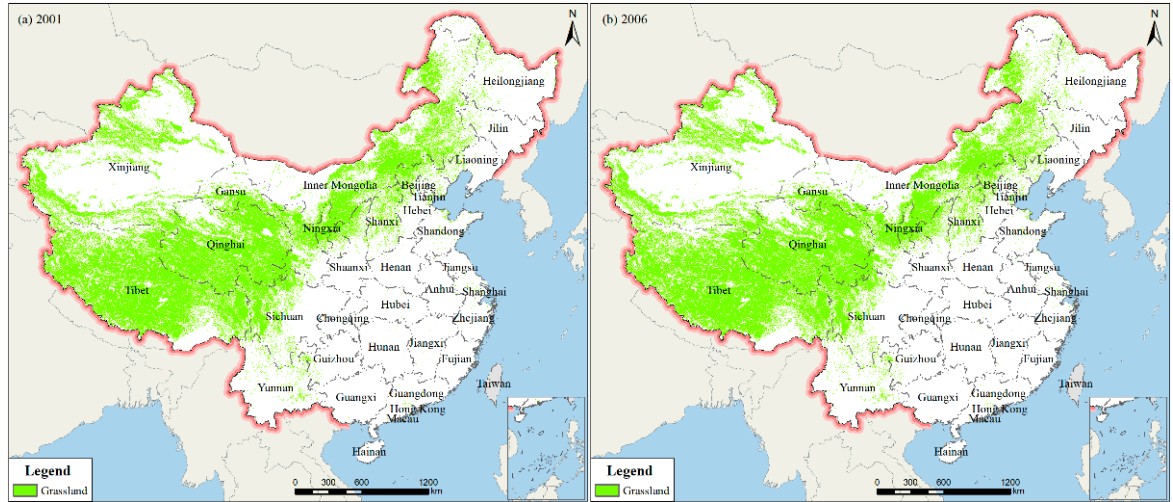

**Figure 4.** *Cont.*

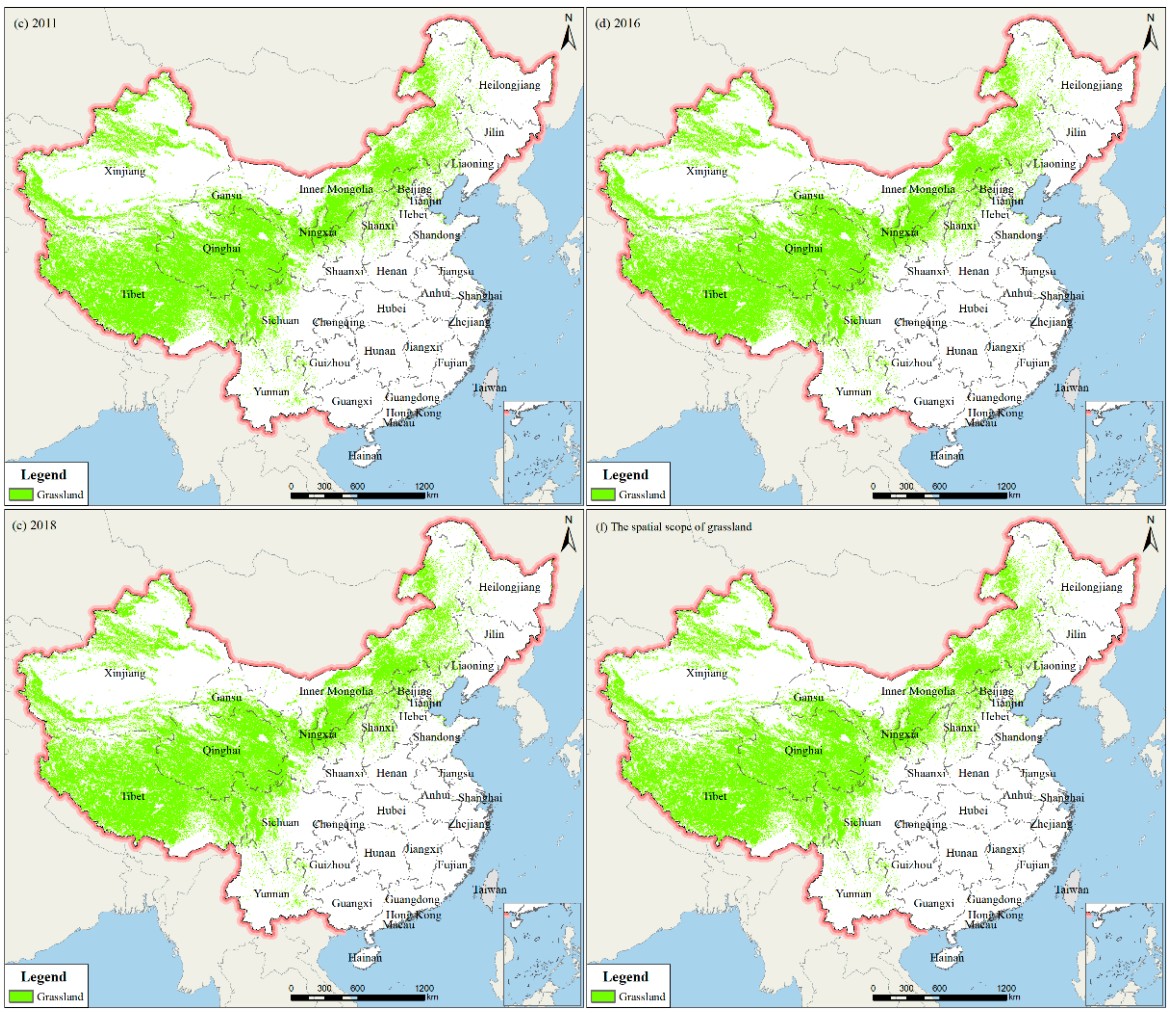

**Figure 4.** The spatial scope of grasslands in China.

### 2.3.2. Calculation of NPP

Improvements to the environmental background-based method were used to calculate the NPP. Three types of NPP were analyzed: actual NPP (*ANPP*), potential NPP (*PNPP*), and human-induced NPP (*HNPP*). The *ANPP* estimates the combined influence of CC and HA. The *PNPP* predicts the ideal condition without human interference and is solely determined by climate conditions. The *HNPP*, representing the influence of HA, can then be calculated using Equation (1):

$$HNPP = PNPP - ANPP \tag{1}$$

The *ANPP* was calculated by NASA, based on MOD17A3 data. For the *PNPP*, since all disturbances are strictly prohibited in the core areas of national nature reserves, the grasslands in the core areas could be assumed to be only affected by CC and free from HA. Therefore, the grassland *PNPP* in the core area was approximately equal to *ANPP*. In addition, if the habitats (altitude, slope, aspect, precipitation, temperature, etc.) of the two pixels were completely similar, we could assume that the *PNPP* of the two pixels could be approximately equal [45], see support information for hypothesis verification. Therefore, the pixel *PNPP* outside the core area could be replaced by the pixel *PNPP* in the nature reserve with the most similar habitat in the same natural ecological geographical division. Similarity pixel matching was mainly based on the principle of the similarity search algorithm of ArcGIS and realized by Matlab programming.

### 2.3.3. Trend Analysis

To analyze the NPP dynamics from 2001 to 2020, the Theil-Sen median slope method was used as the Equation (2) [46,47]. Mann-Kendall (M-K) was utilized to test the significance level of the trend [47].

$$Slope_{sen} = \text{Median}(Q_i) = \text{Median}\left(\frac{x_j - x_i}{j - i}\right) \tag{2}$$

where $x_i$ and $x_j$ are the NPP values in the $i$th and $j$th years ($j > i$), respectively.

### 2.3.4. The Relative Contributions of CC and HA on ANPP

The contribution of CC and HA to NPP could be quantitatively evaluated by the combined slope analysis for the different NPP types [47,48]. A positive $S_{ANPP}$ indicated grassland restoration, while a negative value denoted grassland degradation [10]. Using different $S_{PNPP}$ and $S_{HNPP}$ combinations, six scenarios were identified, characterizing the impact of CC and HA (Table 2) [20].

**Table 2.** Defined scenarios.

| Scheme | | Driving Factors | Contribution | |
|---|---|---|---|---|
| | | | Climate (%) | Human (%) |
| $S_{ANPP} > 0$ | $S_{PNPP} > 0, S_{HNPP} < 0$ | Both | $\frac{100 \times \lvert S_{PNPP}\rvert}{\lvert S_{PNPP}\rvert + \lvert S_{HNPP}\rvert}$ | $\frac{100 \times \lvert S_{HNPP}\rvert}{\lvert S_{PNPP}\rvert + \lvert S_{HNPP}\rvert}$ |
| | $S_{PNPP} < 0, S_{HNPP} < 0$ | Human activities | 0 | 100 |
| | $S_{PNPP} > 0, S_{HNPP} > 0$ | Climate change | 100 | 0 |
| $S_{ANPP} < 0$ | $S_{PNPP} < 0, S_{HNPP} > 0$ | Both | $\frac{100 \times \lvert S_{PNPP}\rvert}{\lvert S_{PNPP}\rvert + \lvert S_{HNPP}\rvert}$ | $\frac{100 \times \lvert S_{HNPP}\rvert}{\lvert S_{PNPP}\rvert + \lvert S_{HNPP}\rvert}$ |
| | $S_{PNPP} > 0, S_{HNPP} > 0$ | Human activities | 0 | 100 |
| | $S_{PNPP} < 0, S_{HNPP} < 0$ | Climate change | 100 | 0 |

Note: $S_{ANPP}$, $S_{PNPP}$, and $S_{HNPP}$ are the slopes of *ANPP*, *PNPP* and *HNPP*, respectively.

### 2.3.5. Correlation Analysis

To identify the correlations between ANPP, temperature, and precipitation, multiple correlation analysis and F-test were used [20]. Partial correlation and t-test were utilized to estimate the relationship of ANPP with temperature and rainfall [49].

## 3. Results

### 3.1. Spatiotemporal Characteristics of Grassland ANPP

#### 3.1.1. Spatial Heterogeneity Analysis of ANPP

The average grassland ANPP from 2001 to 2020 was 148.63 g C·m$^{-2}$·yr$^{-1}$. Regarding the spatial distribution, the grassland ANPP value decreased considerably from the southeast to the northwest of China (Figure 5). The highest ANPP values were mainly distributed in northeastern Inner Mongolia, northeastern Hebei, Shanxi, southeastern Gansu, northeastern Sichuan, northwestern Xinjiang, and some parts of Yunnan (Figure 5).

#### 3.1.2. Spatiotemporal Variation of ANPP

The average grassland ANPP saw a significantly increasing trend (Figure 6), indicating that the synergistic effects of CC and HA led to significant growth of grassland for the given study period. About 89.1% of grassland ANPP exhibited an increasing trend (Figure 7a). The statistically significant increasing trend accounted for 48.3% of grasslands (Figure 7b), mainly distributed in Central Inner Mongolia, Southeastern Qinghai, Northwestern Xinjiang, Northwestern Sichuan, and Northeastern Tibet. In contrast, 10.9% of grassland ANPP showed a declining trend (Figure 7a). Areas witnessing a significantly decreasing trend in ANPP only accounted for 0.6% of grasslands (Figure 7b) and were mainly found in Central and southeastern Tibet, southeastern Qinghai, and northwestern Xinjiang.

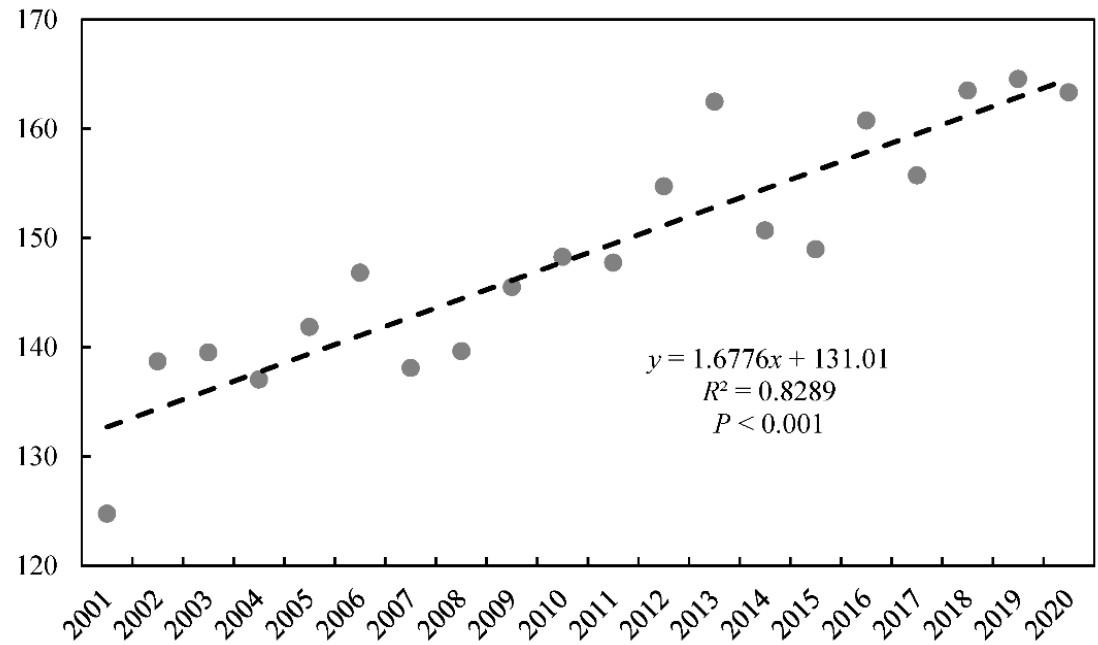

**Figure 5.** Spatial distribution of annual mean ANPP (2001–2020).

$$y = 1.6776x + 131.01$$
$$R^2 = 0.8289$$
$$P < 0.001$$

**Figure 6.** Changes in the annual mean ANPP (2001–2020).

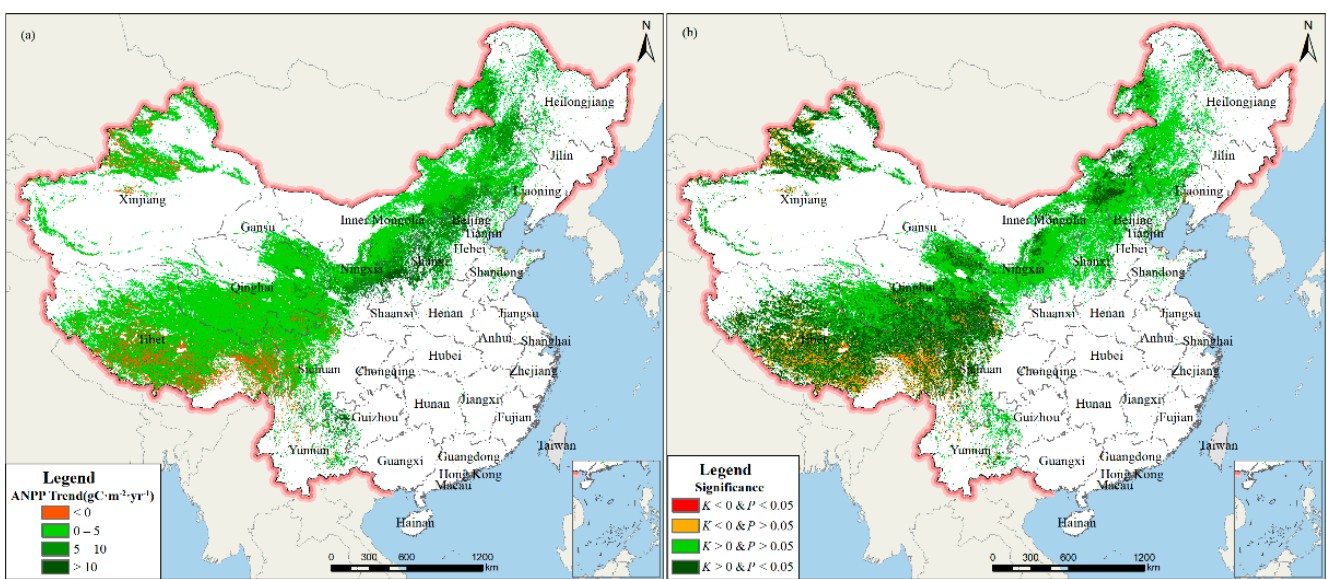

**Figure 7.** (**a**) Trend in ANPP (2001–2020) and (**b**) the significance levels. (Note: *K* is the change trend of ANPP, and *P* is the significance levels).

### 3.2. Contributions of CC and HA to ANPP

3.2.1. Changing Trends of PNPP and HNPP

To differentiate the contributions of CC and HA to ANPP, we first calculated the change trends for PNPP and HNPP from 2001 to 2020. The PNPP trend showed that CC positively affected grassland growth, accounting for 92.9% of the total area. In comparison, only 7.1% of grasslands were adversely affected by CC, mainly situated in southeastern Tibet, Northwestern Xinjiang, Northwestern Sichuan, Northeastern Yunnan (Figure 8).

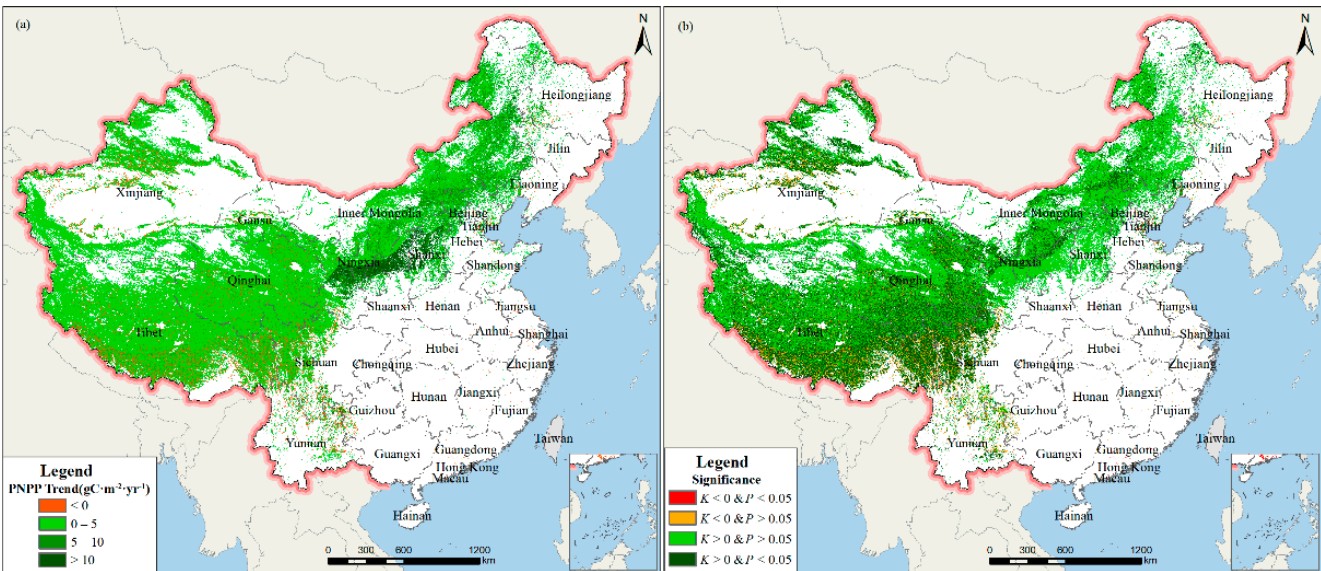

**Figure 8.** (**a**) Trend in PNPP (2001–2020) and (**b**) the significance levels. (Note: *K* is the change trend of ANPP, and *P* is the significance levels).

The HNPP trend suggested that HA had adversely affected grassland growth ($S_{HNPP} > 0$), accounting for 62.0% of the terrestrial area. In the remaining 38.0% of grasslands, anthropogenic activities positively affected grassland growth ($S_{HNPP} < 0$) (Figure 9).

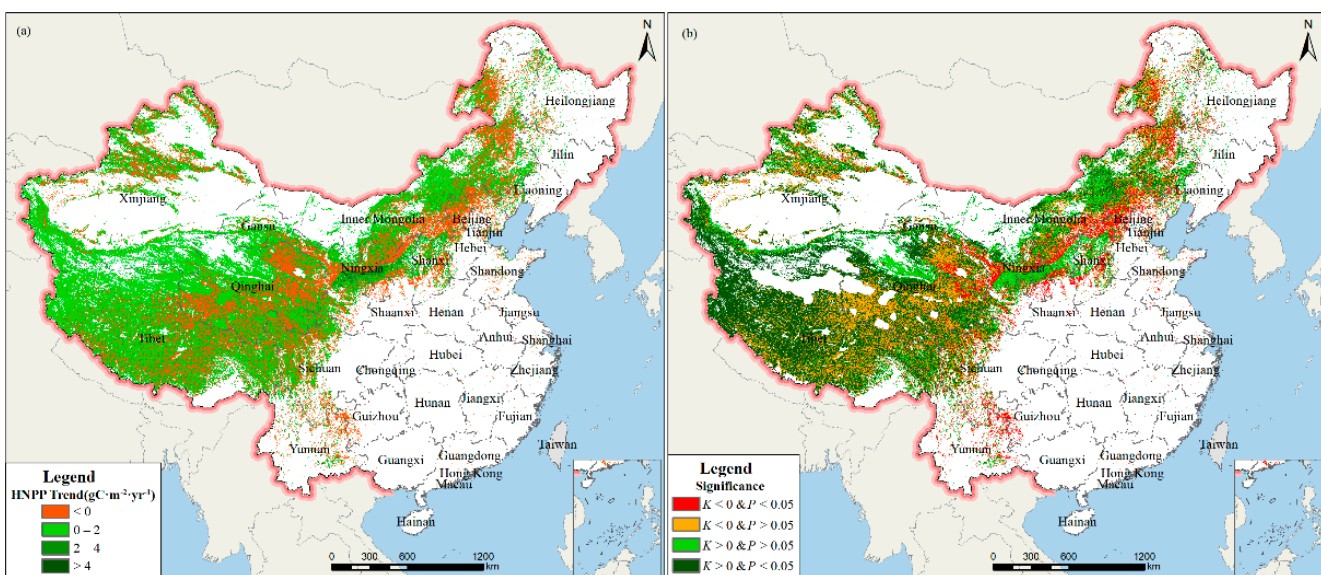

**Figure 9.** (**a**) Trend in HNPP (2001–2020) and (**b**) the significance levels. (Note: *K* is the change trend of ANPP, and *P* is the significance levels).

### 3.2.2. The Relative Contributions of CC and HA to ANPP

Using the calculation results for the ANPP, PNPP, and HNPP change trends, the contribution of CC and HA to ANPP changes were assessed, and the areas that were largely influenced by CC were compared to those primarily affected by HA. Of the 38.4% of the ANPP area affected by CC and HA, climate-dominated areas accounted for 47.8%, whereas human-dominated areas accounted for 13.8% (Figure 10a).

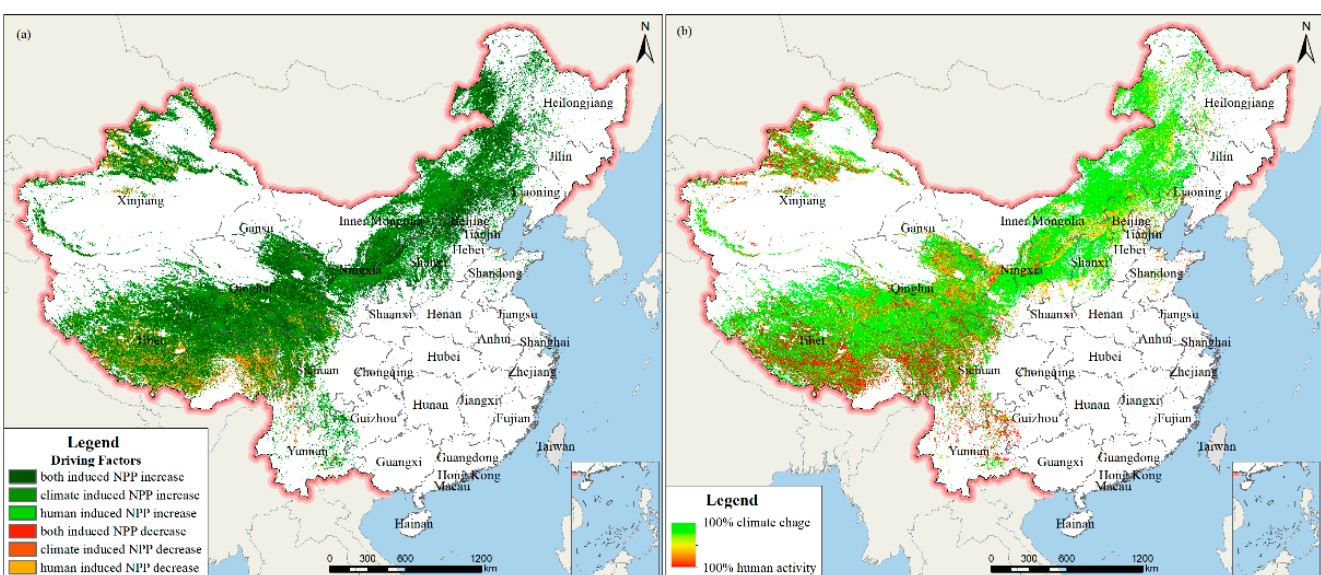

**Figure 10.** Spatial patterns of (**a**) driving factors and (**b**) contributions to ANPP changes.

As shown in Figure 10b, in most of China's grasslands, the influence of CC on ANPP was greater than HA. The few regions where the contributions of HA were greater than that of CC were mainly distributed in Southeast Tibet, Central and Northeastern Qinghai, Western Xinjiang, and Central Inner Mongolia. Generally, the contributions of CC to the changes of ANPP were 71.0%, whereas the effect of HA was 29.0%.

### 3.3. Relationships between ANPP and Climate Factors

### 3.3.1. Multiple Correlations

According to previous research, temperature and precipitation are the dominant climate factors influencing ANPP changes [50–52]. To further explore its internal influence mechanism, a multiple linear regression was applied to estimate the correlation between ANPP and annual average temperature and annual average total precipitation. The results showed that ANPP was positively correlated with temperature and precipitation. Those areas with a significant positive correlation accounted for 46.5% of the total area. In comparison, the remaining 53.5% that did not exhibit a significant positive correlation were mainly distributed in Northwestern Xinjiang, Northwestern Sichuan, Southeastern Tibet, Southeastern Qinghai, and Northern Shaanxi (Figure 11). We also further estimated the correlation between ANPP and the average temperature and total precipitation in the growing season (Figure 12), and the spatial distribution of correlation coefficient was basically consistent with Figure 11. Those areas with a significant positive correlation accounted for 40.5% of the total area. In comparison, the remaining 59.5% did not exhibit a significant positive correlation (Figure 12).

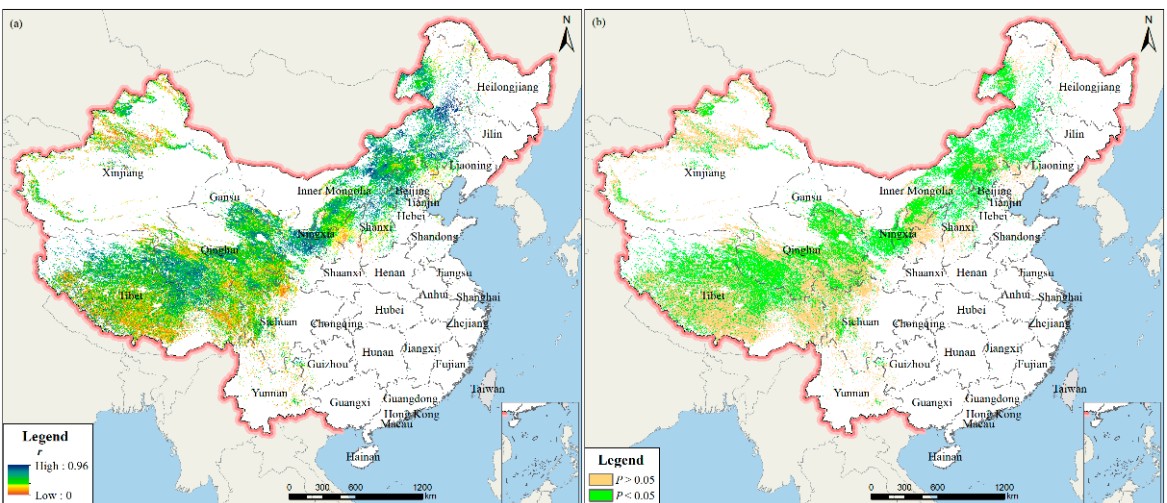

**Figure 11.** Spatial distribution of (**a**) multiple correlation coefficient between ANPP and annual average temperature and annual average total precipitation and (**b**) the significance level.

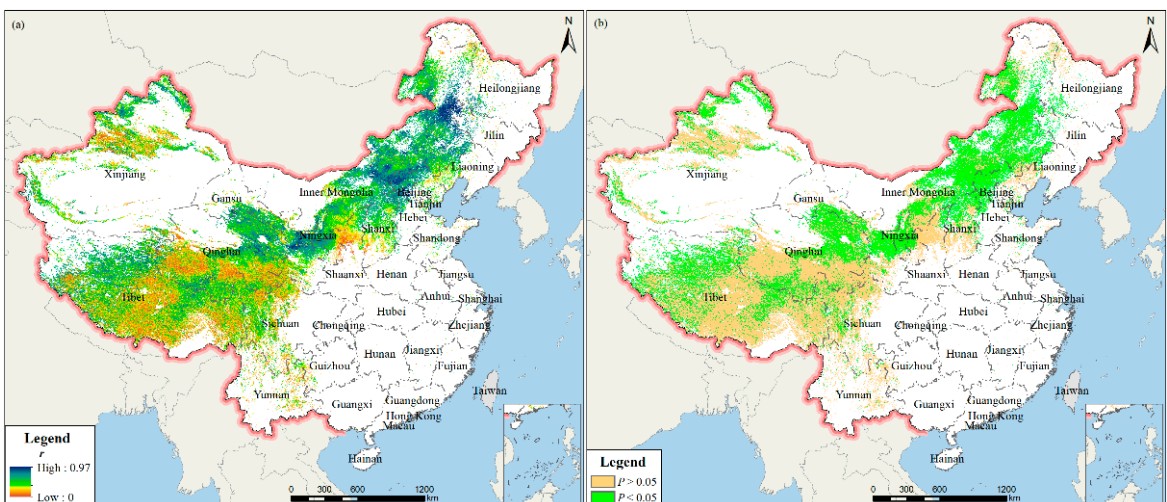

**Figure 12.** Spatial distribution of (**a**) multiple correlation coefficient between ANPP and the average temperature and total precipitation in the growing season and (**b**) the significance level.

### 3.3.2. Partial Correlations

To spatially determine the relationship of ANPP with annual average temperature and annual average total precipitation, we mapped their partial correlation results (Figures 13 and 14). Precipitation was positively correlated with ANPP for 67.0% of the grassland areas. Areas with significantly positive relationships were mainly distributed in Ningxia, Southeast Qinghai, Inner Mongolia, and Northwest Xinjiang, accounting for 30.5% (Figure 13). Temperature was positively correlated with ANPP in 73.1% of the grasslands, and regions with significant positive correlations were mainly distributed in Qinghai, Tibet, Central Gansu, and Central Ningxia, accounting for 26.9% (Figure 14). We also further estimated the partial correlation between ANPP and the average temperature and total precipitation in the growing season (Figures 14 and 15). Precipitation was positively correlated with ANPP for 66.2% of the grassland areas. Areas with significantly positive relationships were mainly distributed in Ningxia, Southeastern Qinghai, Inner Mongolia, and Northwest Xinjiang, accounting for 28.3% (Figure 15). Temperature was positively correlated with ANPP in 62.0% of the grasslands, and regions with significant positive correlations were mainly distributed in Qinghai, Tibet, Central Gansu, and Central Ningxia, accounting for 38.0% (Figure 16).

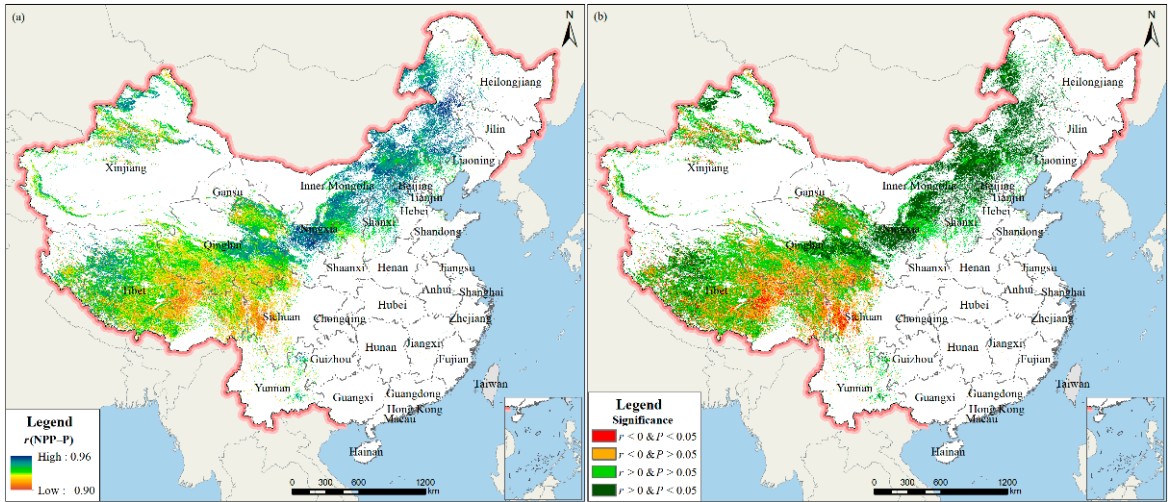

**Figure 13.** Spatial patterns of (**a**) partial correlation coefficients between ANPP and annual average total precipitation and (**b**) the corresponding significance levels.

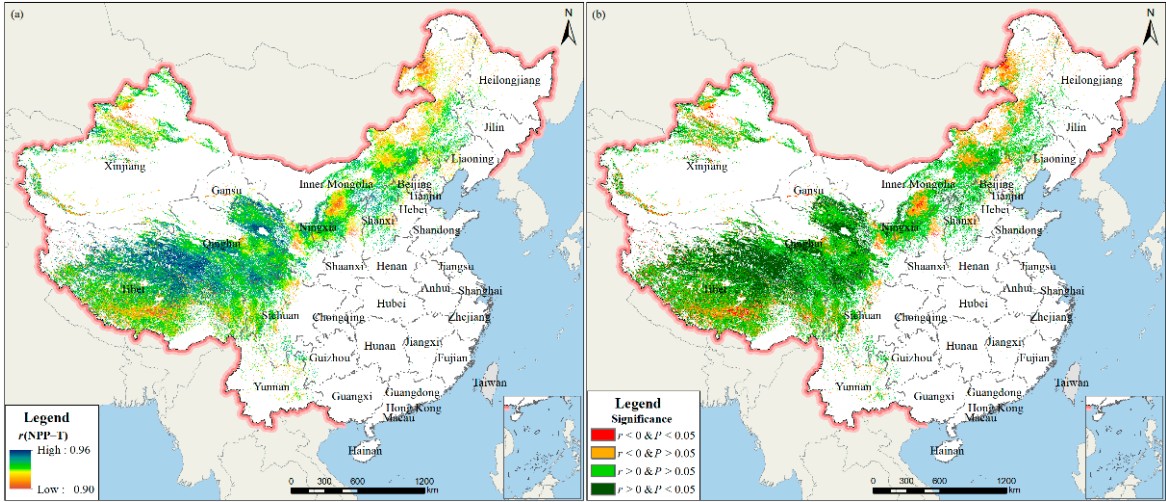

**Figure 14.** Spatial patterns of (**a**) partial correlation coefficients between ANPP and annual average temperature and (**b**) the corresponding significance levels.

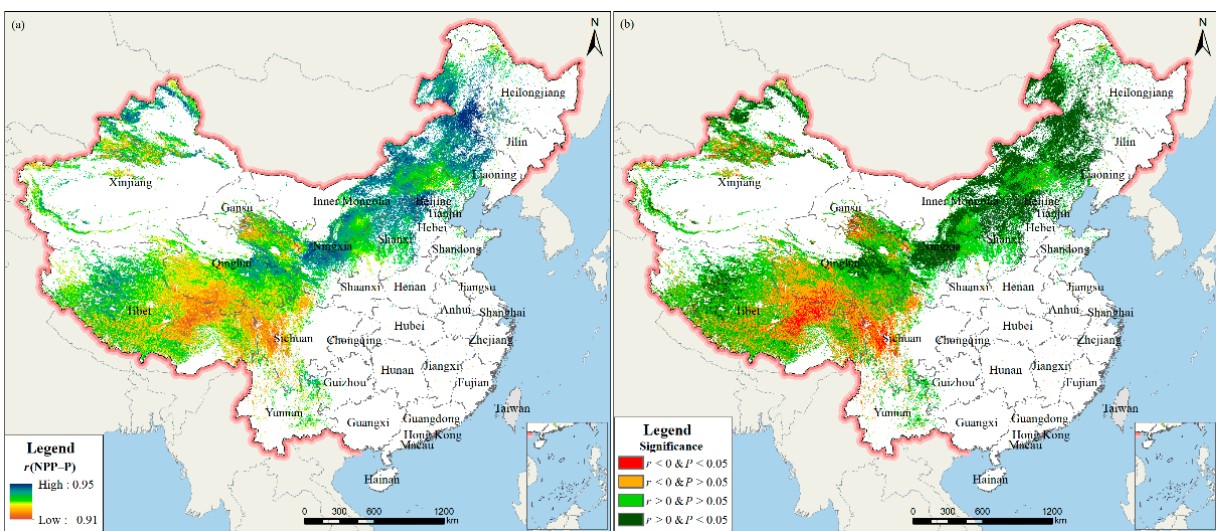

**Figure 15.** Spatial patterns of (**a**) partial correlation coefficients between ANPP and total precipitation in the growing season and (**b**) the corresponding significance levels.

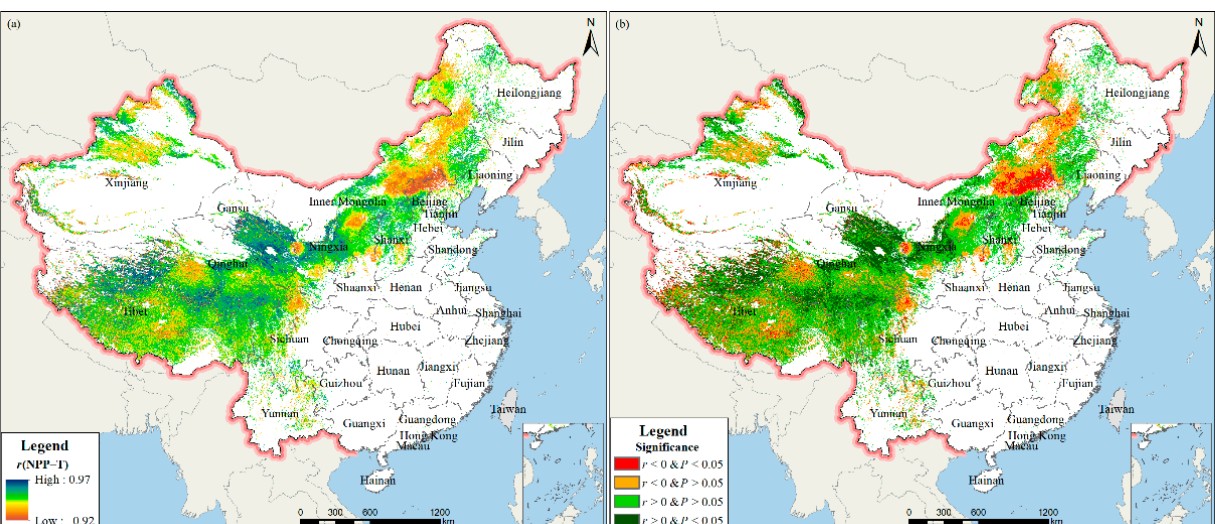

**Figure 16.** Spatial patterns of (**a**) partial correlation coefficients between ANPP and the average temperature in the growing season and (**b**) the corresponding significance levels.

## 4. Discussion

### 4.1. Effects of CC on ANPP

Climate change is a crucial determinant in vegetation dynamics [53–55]. Numerous studies have identified precipitation and temperature as the primary climate factors influencing ANPP changes [56,57]. Based on the multiple and partial correlation analyses between ANPP and the main climate factors, we found that the ANPP of China's grasslands generally showed an increasing trend over the last 20 years. Precipitation was found to be conducive to grassland growth, particularly in the arid and semi-arid regions, such as Gansu, Inner Mongolia, and Xinjiang (Figures 13 and 15). In contrast, our results showed that temperature hindered grassland growth in these areas (Figures 14 and 16), consistent with the conclusions of previous studies [1]. Since water is the most dominant factor affecting grassland productivity in arid and semi-arid areas, temperature increase may lead to the intensification of evaporation and water reduction, leading to increased grassland degradation [31,51].

This study assessed the CC and HA influences on changes in grassland ANPP. Overall, CC positively affected ANPP in most of China's grasslands and had a greater effect than HA (Figure 10). This finding is consistent with previous studies [1]. Our results also indicated that CC was not conducive to grassland ANPP changes in some regions, particularly in southeastern Tibet (Figure 10). This was mainly attributed to the decrease in precipitation and the increase in temperature in the area during this period, considerably hindering grassland growth [31].

In addition, we found the contribution of CC was less in grassland restoration areas, such as Central Qinghai, Central Inner Mongolia, Central Tibet and Western Xinjiang (Figure 10). Among them, the contribution of CC in Central Qinghai and Central Inner Mongolia was small, mainly because they were located in population concentration areas, and the contribution of human activities was large, and had a positive impact on grassland restoration, which was consistent with [31,32]. However, the contribution of CC in Central Tibet and other places was small, mainly due to the obvious weakening of grazing activities in the past 20 years [3]. The contribution of CC in western Xinjiang and other places was small, and inconsistent with [31,32], which might have been due to inconsistent methods.

### 4.2. Effects of HA on ANPP

Anthropogenic activities are also critical to grassland NPP changes [5,52,58]. Human actions, for instance, overgrazing, land use change, and excessive exploitation, can lead to significant grassland degradation [9,59]. Our results showed that HA was the main cause of degradation in China's grasslands (Figure 10). The degraded areas were mainly located in Southeastern Tibet, Northwestern Xinjiang, and Southeastern Qinghai (Figure 10), and this was mainly due to the high grazing intensity in these areas [10,31]. This result is consistent with [31,36].

In most grassland areas in China, HA adversely affected grassland growth ($S_{HNPP} > 0$), accounting for 62.0% of the terrestrial area (Figure 9). The statistically significant increasing trend accounted for 15.1% of grasslands, mainly distributed in Central Qinghai, Southwestern Xin-jiang, and Southwestern Tibet. Among them, HA in Southwestern Xinjiang and Southwestern Tibet had a great impact, mainly due to the increased grazing activities [10,31]. HA in central Qinghai had a great impact, mainly because it was located in a population concentration area.

### 4.3. Methods for Quantitative Assessment of the CC and HA Influence on Grassland ANPP

The quantitative and comprehensive evaluation of climate change effects and anthropogenic influences on the NPP of grasslands remains a major challenge [60,61]. Previous studies have largely analyzed the effects of CC or HA on grassland NPP separately [62–65], with only a few studies conducting comprehensive assessments. The most widely used quantitative methods (i.e., regression analysis, residual trend, biophysical model, and environmental background-based methods) all contain considerable levels of uncertainty [1,24,66]. To improve the accuracy of the grassland NPP assessment, we introduced modifications to the environmental background-based method to calculate the contributions of CC and HA to NPP.

In our method, grassland NPP in nature reserves is used as the environmental background value and was assumed to be affected only by CC and free from HA. The impacts of anthropogenic activities on vegetation NPP were then characterized by calculating the difference between the environmental background value and the NPP outside the reserve. Our results were largely consistent with [31,34], suggesting the reliability of the developed approach.

Our proposed method is superior to the regression analysis and the residual trend methods since it improves the differentiation of climatic and anthropogenic effects on grassland NPP. Our method is comparatively better than the biophysical model-based approach because it can avoid calculation uncertainties from the various model parameters. Moreover, our method provides better results compared to the traditional environmental

background-based approach because the impact of eco-geographical zoning on grassland NPP can be removed. Therefore, this research could provide support for government decisions and methodological reference for the quantitative evaluation of grasslands and other ecosystems.

However, there are also some limitations to this study. First, we assumed that the grassland PNPP pixels outside the core area could be replaced by the pixel PNPP in the nature reserve with the most similar habitat in the same natural ecological geographical division. Due to the availability of data, we only judged the critical factors, such as precipitation, temperature, slope, aspect and elevation, as similar habitat conditions, ignoring the impact of biotic factors (e.g., introduced species, succession) [67]. There were certain errors in some areas where natural biotic factors interfered greatly. Second, although the method constructed in this research accurately identified the contributions of CC and HA to grassland NPP, it could not further separate the impact of different types of human activities (e.g., overgrazing or environmental pollution) on net primary productivity. Therefore, we will focus on solving the above problems in future research.

### 5. Conclusions

In this study, an improved environmental background-based method was developed to differentiate the influence of CC and HA on grassland NPP. Our results showed that the average grassland ANPP in China from 2001 to 2020 showed a significant increasing trend, with 89.1% of grasslands showing a rising trend, while 10.9% exhibited a decreasing trend. The findings also suggested that CC had a more considerable effect on grassland NPP changes than HA in China's grasslands. CC contributed 71.0% to ANPP change, whereas HA contributed 29.0% to ANPP change. Precipitation was found to be conducive to grassland growth among arid and semi-arid regions, whereas temperature inhibited grassland growth in these areas. While HA was the main cause of grassland degradation, its effects on grasslands have decreased in most of China's grasslands.

**Author Contributions:** Conceptualization, F.Y. and X.Z.; methodology, F.Y., X.-C.W., X.Z. and Y.Z.; software, B.P. and X.Z.; validation, B.P.; analysis, X.Z.; investigation, X.Z.; writing—original draft preparation, X.Z.; writing—review and editing, F.Y. and X.-C.W.; funding acquisition, F.Y. and X.-C.W. All authors have read and agreed to the published version of the manuscript.

**Funding:** This research is supported by the National Natural Science Foundation of China (72074156), the Major Project of the National Social Science Foundation of China (21&ZD166), Fundamental Research Funds for the Central Universities (310421102), and the State Key Laboratory of Earth Surface Processes and Resource Ecology project (370100014).

**Data Availability Statement:** The data presented in this study can be available on request from the corresponding author.

**Acknowledgments:** The authors would like to thank USGS and Resource and Environment Science and Data Center for providing the research data, and also thank Yu and Wang for patient guidance. The authors are also grateful to the editor and anonymous reviewers for their positive comments on the manuscript.

**Conflicts of Interest:** The authors declare no conflict of interest.

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
