# Peer review of "Quantifying the Influence of Climate Change and Anthropogenic Activities on the Net Primary Productivity of China’s Grasslands"

_remotesensing, doi:10.3390/rs14194844_

Round 1

Reviewer 1 Report (Previous Reviewer 1)

Thank you for the revision. In the discussion section regarding study limitations, please consider fully elaborating on the possible effects of excluding human activity types and natural biotic factors from the analysis on the results.

Author Response

Dear Editor and Reviewers,

Thank you very much for handling and reviewing the manuscript. Your comments and suggestions are so valuable and helpful for revising and improving this work. All of them have been carefully studied. Responses/explanations have been provided. Revisions have been made. The main corrections in the paper and the responses to the comments are as followings. We hope they can meet your requirements.

Response to the Comments from Reviewer #1

Reviewer #1: Thank you for the revision. In the discussion section regarding study limitations, please consider fully elaborating on the possible effects of excluding human activity types and natural biotic factors from the analysis on the results.

Response: Thank you very much for the comment. We agree with you. We supplemented relevant discussions in Section 4.3, see the revised paper for details.

Reviewer 2 Report (Previous Reviewer 3)

Dear Editor,

Please find my review of a study "Quantifying the influence of climate change and anthropogenic activities on the net primary productivity of China’s grasslands" by Xiafei Zhou , Binbin Peng , Ying Zhou , Fang Yu, Xue-chao Wang submitted to Remote Sensing.

This study aims to improve assessment accuracy in quantifying the contributions of climate change and human activities on grassland net primary productivity in China. To calculate contributions of climate change and human activities to the actual net primary productivity, modifications to the environmental background-based method were introduced. It was found that climate change contributed about two thirds to the actual net primary productivity change.

The subject of this report is suitable for Remote Sensing and it could be published in this journal.

I reviewed the original version of the manuscript in June 2022 and provided a number of comments. The authors have revised the manuscript addressing my comments. This reviewer recommends accepting the manuscript for publication.

Yours faithfully,

The Reviewer

Author Response

Dear Editor and Reviewers,

Thank you very much for handling and reviewing the manuscript. Your comments and suggestions are so valuable and helpful for revising and improving this work. All of them have been carefully studied. Responses/explanations have been provided. Revisions have been made. The main corrections in the paper and the responses to the comments are as followings. We hope they can meet your requirements.

Response to the Comments from Reviewer #2

Reviewer #2: This study aims to improve assessment accuracy in quantifying the contributions of climate change and human activities on grassland net primary productivity in China. To calculate contributions of climate change and human activities to the actual net primary productivity, modifications to the environmental background-based method were introduced. It was found that climate change contributed about two thirds to the actual net primary productivity change.

The subject of this report is suitable for Remote Sensing and it could be published in this journal.

I reviewed the original version of the manuscript in June 2022 and provided a number of comments. The authors have revised the manuscript addressing my comments. This reviewer recommends accepting the manuscript for publication.

Response: Thank you very much for reviewing the paper.

This manuscript is a resubmission of an earlier submission. The following is a list of the peer review reports and author responses from that submission.

Round 1

Reviewer 1 Report

Thank you for the invitation to review this submission. Understanding how natural and anthropogenic factors influence grassland productivity is definitely an important and interesting research topic. The manuscript is also well written in general. But I'm afraid I can't support publishing the study, mostly because I have a lot of doubts about how the experiment was set up.

First of all, to my understanding, the validity of the whole analysis is based on the assumption that "Within the same eco-geographical division, the grassland NPP in different regions is assumed to be roughly equal." Since various local factors (e.g., elevation, slope, aspect, disturbance history, local community composition, etc.) could affect changes in grassland productivity and the eco-geographical zoning is very large-scale, this crucial assumption definitely needs to be tested. Without solid evidence, I have strong concerns regarding the accuracy of the results. The "matching" method might be considered to find suitable pixels outside of nature reserves that could be used as "treatments" to compare the NPP differences.

In addition, in terms of the contribution of climate and human activities to NPP, the authors only used annual mean temperature and precipitation (I assumed because I didn’t find clear descriptions) as the proxies for climate factors and no specific factors for human activities. However, since climate conditions other than the annual means (e.g., different timing of precipitation and temperature during growing seasons, etc.) and different types of human activities (e.g., urbanization, grazing, etc.) could also significantly affect the productivity of ecosystems in different ways, much more important questions that need to be answered should be like what exact climate conditions and types of human activity affected the changes in NPP. Moreover, I am also wondering how the changes in natural biotic factors (e.g., introduced species, succession), which might not be considered as climate and anthropogenic factors, affect the changes in NPP. Last but not least, more robust slope calculation methods such as the Theil-Sen estimator are also suggested for trend detection due to the high uncertainties in the modelled NPP products.

I also have some minor comments, but these major areas of concern weigh more heavily on my mind in reviewing the study.

Reviewer 2 Report

Dear Authors,

The submitted manuscript titled „Quantifying the influence of climate change and anthropogenic activities on the net primary productivity of China’s grasslands” contains interesting results, which might interest the international audience. In my opinion manuscript is generally well-planed, however I have some suggestions of improvements and questions worth solving. Please, find them below:

1.       In my opinion the grasslands should be described more detaildly. I think that Figure 1 is too general. It is worth to distinguish the type of grasslands. Information about type of management should be added. Perhaps publication of Kang et al. (2007) would be useful.

Kang L, Han X, Zhang Z, Sun OJ. Grassland ecosystems in China: review of current knowledge and research advancement. Philos Trans R Soc Lond B Biol Sci. 2007 Jun 29;362(1482):997-1008.

2.       In my opinion the concise presentation of current state of knowledge on main study topic should be very interesting to readers. Moreover, it will  justify the undertaking of research.

3.       Are there the differences among type of grasslands in investigated parameters?

4.       The Figures, Tables and charts should be self-explanatory, therefore all abbreviations should be explained in captions. Also axa in charts should be described.

5.       The valuable results are poorly discussed. Taking into account the voluminous literature of subject the chapter Discussion should be enlarged. In my opinion the comparisons with similar investigations contucted in other regions (climate zones) would be interesting.

Perhaps some of below listed literature sources might be useful in manuscript improvemets:

·         Tianjie Lei, Jianjun Wu, Xiaohan Li, Guangpo Geng, Changliang Shao, Hongkui Zhou, Qianfeng Wang, Leizhen Liu 2015. A new framework for evaluating the impacts of drought on net primary productivity of grassland. Science of The Total Environment 536: 161-172. https://doi.org/10.1016/j.scitotenv.2015.06.138.

·         Heisler-White, J.L., Knapp, A.K. & Kelly, E.F. 2008. Increasing precipitation event size increases aboveground net primary productivity in a semi-arid grassland. Oecologia 158, 129–140.

·         Ni, J. Estimating net primary productivity of grasslands from field biomass measurements in temperate northern China. 2004. Plant Ecology 174, 217–234. https://doi.org/10.1023/B:VEGE.0000049097.85960.10

·         Shaojie Mu, Shuangxi Zhou, Yizhao Chen, Jianlong Li, Weimin Ju, I.O.A. Odeh. 2013. Assessing the impact of restoration-induced land conversion and management alternatives on net primary productivity in Inner Mongolian grassland, China,         Global and Planetary Change 108, 29-41. https://doi.org/10.1016/j.gloplacha.2013.06.007.

·         Chengcheng Gang, Wei Zhao, Ting Zhao, Yi Zhang, Xuerui Gao, Zhongming Wen 2018. The impacts of land conversion and management measures on the grassland net primary productivity over the Loess Plateau, Northern China. Science of The Total Environment 645: 827-836. https://doi.org/10.1016/j.scitotenv.2018.07.161.

Reviewer 3 Report

Dear Editor,

Please find my review of a study "Quantifying the influence of climate change and anthropogenic activities on the net primary productivity of China’s grasslands" by Xiafei Zhou , Binbin Peng , Ying Zhou , Fang Yu, Xue-chao Wang submitted to Remote Sensing.

This study aims to improve assessment accuracy in quantifying the contributions of climate change and human activities on grassland net primary productivity in China. To calculate contributions of climate change and human activities to the actual net primary productivity, modifications to the environmental background-based method were introduced. It was found that climate change contributed about two thirds to the actual net primary productivity change.

The subject of this report is suitable for Remote Sensing and it could be published after suggested revision.

Abstract.

Precipitation is decisive in grassland growth among arid and semi-arid regions, while temperature inhibited productivity in these areas. => Precipitation was the main contributor to grassland growth among arid and semi-arid regions, while temperature inhibited productivity in these areas.

Introduction.

CC (e.g., rainfall and temperature change) will significantly affect the intensity of vegetation photosynthesis and respiration [11, 12], leading to an increase or decrease in vegetation NPP [7, 13]. => CC (e.g., precipitation and temperature change) will significantly affect the intensity of vegetation photosynthesis and respiration [11, 12], leading to an increase or decrease in vegetation NPP [7, 13].

However, due to climate warming, land-use changes, and overgrazing, many grasslands in China have suffered serious ecological degradation [10, 31], causing significant declines in the regional carbon sequestration capacity [32, 33]. => However, due to climate and land-use changes, as well as overgrazing, many grasslands in China have suffered serious ecological degradation [10, 31], causing significant declines in the regional carbon sequestration capacity [32, 33].

2. Materials and methods

2.2. Data sources

The land use data for 2000-2018, the net primary productivity data for 2001-2020 and precipitation and temperature during 2001-2020 were obtained from the European Space Agency, the United States Geological Survey and the China Meteorological Data Network, respectively.      Data sources should be mentioned more specifically, e.g., links to websites with open access data from the European Space Agency and the United States Geological Survey should be included.

2.3.2. Calculation of NPP

The HNPP, representing the influence of HA, can then be calculated using the following equation => The HNPP, representing the influence of HA, can then be calculated using the Equation 1

2.3.3. Trend analysis

To analyze the NPP dynamics from 2001 to 2020, the ordinary least squares method was used as the following formula =>  To analyze the NPP dynamics from 2001 to 2020, the ordinary least squares method was used as the Equation 2

4. Discussion

4.1. Effects of CC on ANPP

Precipitation was found to be conductive to grassland growth, particularly in the arid and semi-arid regions, such as Gansu, Inner Mongolia, and Xinjiang (Fig. 12). => Precipitation was found to be conducive to grassland growth, particularly in the arid and semi-arid regions, such as Gansu, Inner Mongolia, and Xinjiang (Fig. 12).

4.2. Effects of HA on ANPP

This result is consistent with Zhou et al, Yan et al [31, 36]. => This result is consistent with [31, 36].

4.3. Methods for quantitative assessment of the CC and HA influence on grassland ANPP

Our results are largely consistent with Zhou et al, Yan et al [31, 34], suggesting the reliability of the developed approach. => Our results are largely consistent with [31, 34], suggesting the reliability of the developed approach.

To conclude, this reviewer recommends minor revision.

Yours faithfully,

The Reviewer

Round 2

Reviewer 1 Report

Thank you for the revision and reply. However, none of my major concerns have been addressed substantially through any additional analysis and provision of supporting evidence. Therefore, I am afraid that I must maintain my previous decision.

Reviewer 2 Report

Dear Authors,

Thank You for explanations. I do not have other suggestions and remarks for further corrections.